



# Drastic changes in Depositional Environments at the Ross Sea Continental Margin since the Mid-Pleistocene: More evidence for West Antarctic Ice Sheet collapse

Chinmay Dash[1] , Yeong Bae Seong[1], Ajay Kumar Singh[2], Min Kyung Lee[3], Jae Il Lee[3], Kyu-Cheul Yoo[3],
Hyun Hee Rhee[4], Byung Yong Yu[5]

[1]Department of Geography Education, Korea University, Seoul 02841, Korea
[2]Independent Researcher, Samastipur, Bihar 848101, India
[3]Division of Glacial Environment Research, Korea Polar Research Institute, Incheon 21990, Korea
[4]Department of Environment Education, Kongju National University, Chungnam 32588, Korea
[5]AMS Laboratory, Korea Institute of Science and Technology, Seoul 02792, Korea

*Correspondence to*: Yeong Bae Seong (ybseong@korea.ac.kr)

**Abstract.** This study investigates a sediment core (RS15-LC47) from the Ross Sea continental rise to elucidate the sea-ice interaction and resulting paleodepositional changes over the past 800 ka. By integrating whole-core Magnetic Susceptibility (MS), sediment biogenic components (TOC, $CaCO_3$, and biogenic silica), sedimentological features, and the isotopic ratio of authigenic beryllium $(^{10}Be/^9Be)_{reac}$, we unravel the paleoenvironmental changes and their influence on the sedimentary processes. The lower segment of the investigated interval (750-550 ka) exhibits distinct lithological characteristics, including parallel and cross laminations, along with millimeter-scale faults, suggestive of contourite depositional processes. This section also displays irregular trends in MS values due to poorly sorted sediments, characteristics feature of sediment slumping. The lowest $(^{10}Be/^9Be)_{reac}$ ratio in this interval suggests reduced Circumpolar Deep Water (CDW) inflow due to strengthened Antarctic Slope Current (ASC). Although the Total Organic Carbon (TOC) is highest in this interval, high Carbon-to-Nitrogen (C/N) ratio and low Barium excess $(Ba_{ex})$ suggests reduced marine productivity due to increased terrestrial input likely from advancing ice sheets. Following the Mid-Pleistocene Transition (MPT), MS values remain consistently low until MIS 8 (~250 ka) and $(^{10}Be/^9Be)_{reac}$ relatively increases, indicating persistent lukewarm condition. We hypothesize this timeframe favourable for ice-shelf disintegration and possible collapse of the West Antarctic Ice Sheet (WAIS). Between 550 and 250 ka, TOC/TN levels resemble those observed in the euphotic layer of the Ross Sea, with relatively higher $Ba_{ex}$ and TOC, indicating higher productivity during an extended lukewarm condition. During the late Pleistocene (> 250 ka), coarser grain size and IRD-rich layers suggest strengthening of bottom currents. The upwelling of CDW facilitated a drastic increase in the $(^{10}Be/^9Be)_{reac}$ ratio during the late Pleistocene. $Opal_{MAR}$ and TOC % exhibit positive trends with $(^{10}Be/^9Be)_{reac}$ during the late Pleistocene interglacials, suggesting increased productivity during warmer periods.

**Short summary.** This study explores sediment core RS15-LC47 from the Ross Sea over the past 800,000 years, examining changes in sea-ice dynamics and deposition environments. It integrates various data to reveal shifts related to Circumpolar



Deep Water influx and Antarctic currents, particularly during significant climate transitions. Results highlight potential West Antarctic Ice Sheet collapses in warmer periods, offering new insights into the area's paleoclimate and sedimentary processes.




## 1 Introduction

The Ross Sea contributes about 40% of the total Antarctic Bottom Water (AABW) volume (Orsi et al., 2002) and is the primary drainage outlet of grounded ice from both eastern and western Antarctica. Empirical models have highlighted the susceptibility

of sea ice extent in the Ross Sea to the ocean–atmospheric interactions and the changes in the heat flux during the glacial-interglacial cycles (Yuan et al., 2017). Thus, studying the sedimentation pattern in this region is critical in elucidating paleoceanographic changes and historical ice sheet dynamics. Recent geological investigations underscore the significance of comprehending past interactions between the ocean and ice sheets in the Ross Sea sector (Nishimura et al., 1998; Jacobs et al., 2002; Chow and Bart, 2003; Bart and Owolana, 2012; McKay et al., 2012b; Halberstadt et al., 2016; Anderson et al., 2019;

McKay et al., 2019; Silvano et al., 2020), particularly in light of the retreating West Antarctic Ice Sheet (WAIS) (Pattyn and Morlighem, 2020). The ANDRILL 1-B core revealed 28 glacial-interglacial cycles between 5 to 1 Ma, with several hiatuses in sediment record; probably representing periods of glacial erosion (Naish et al., 2009). Although the evidence for the disintegration of WAIS remains uncertain during the past 1 Ma, glacial-interglacial facies cyclicity recorded in the AND-1B sediment core suggests episodic changes from floating to grounded state of Ross Ice shelf. Such changes seem to have been

occurring in tandem with orbital cyclicity (McKay et al., 2012b).

Furthermore, the Ross Sea sector experienced drastic oceanographic changes during-and after- Mid-Pleistocene Transition (MPT), significantly affecting sedimentation across the continental margin. For instance, Konfirst et al. (2012) suggested the expansion of the Ross Gyre prior to MIS 16 in response to the deepening of the Amundsen Sea low, which transported the ice-proximal deposits to the outer sea (Wang et al., 2022). Organic geochemical and sedimentary facies records from AND-1B

showed a consistent increase in the duration and the coverage of sea ice during the past 3 Ma (Naish et al., 2009) and significant ice expansion during MPT (McKay et al., 2012b). Nevertheless, the majority of research in this sector of Antarctica regarding the ice sheet variation is restricted to the last few hundred thousand years because of harsh conditions and challenges in establishing chronological records from sedimentary deposits. Consequently, there is a notable scarcity of continuous paleoenvironmental records in this specific part of the Antarctic continental margin. Most of the analyzed sediment cores in

the Ross Sea are from the continental shelf region, where glacial overriding disrupts the continuity of the stratigraphic record. Consequently, exploring sediment records from the continental slope and rise could offer a continuous paleoenvironmental record.

 We investigated a sediment core (RS15-LC47) from the Ross Sea continental rise to comprehend regional paleoceanographic changes since the mid-Pleistocene (Fig. 1). We employed whole core Magnetic Susceptibility (MS), sediment biogenic

components (TOC, CaCO3 and Biogenic silica), sedimentological characteristics and Beryllium (Be) isotopes in reactive phases ($^{10}Be_{reac}$, $^{9}Be_{reac}$, and $^{10}Be_{reac}/^{9}Be_{reac}$) to decipher the paleodepositional environment. While sedimentological and organic geochemical parameters have conventionally been employed to comprehend depositional environments in the



Southern Ocean region, the application of meteoric $^{10}$Be has been limited to a few studies (Frank et al., 2002; Chase et al., 2003; Sjunneskog et al., 2007; Yokoyama et al., 2016; Valletta et al., 2018; Dash et al., 2021; Rhee et al., 2022). Meteoric

$^{10}$Be has been widely used in glaciomarine sediments of the Southern Ocean for tracing depositional processes and assessing the dynamics of Antarctic ice sheets/ice shelves (Yokoyama et al., 2016; Valletta et al., 2018). However, to accurately capture the history of glacial retreat and depositional processes, it's imperative to consider that the meteoric $^{10}$Be signal can be influenced by other factor(s) beyond environmental forces alone, such as particle size (Kretschmer et al., 2011; Wittmann et al., 2012). To address these issues, researchers have turned to measuring "reactive" $^{10}$Be ($^{10}$Be$_{reac}$) bound to Fe and Mn

oxyhydroxides, which scavenge meteoric $^{10}$Be from the water column (von Blanckenburg et al., 2012), in conjunction with its stable counterpart $^{9}$Be in reactive phase ($^{9}$Be$_{reac}$), derived from bedrock weathering (Wittmann et al., 2012). Our objective is to generate more comprehensive records of paleoenvironmental changes and their impact on the depositional environment at the Antarctic continental margin by combining $^{10}$Be$_{reac}$, ($^{10}$Be/$^{9}$Be)$_{reac}$, sedimentological and organic geochemical proxies.

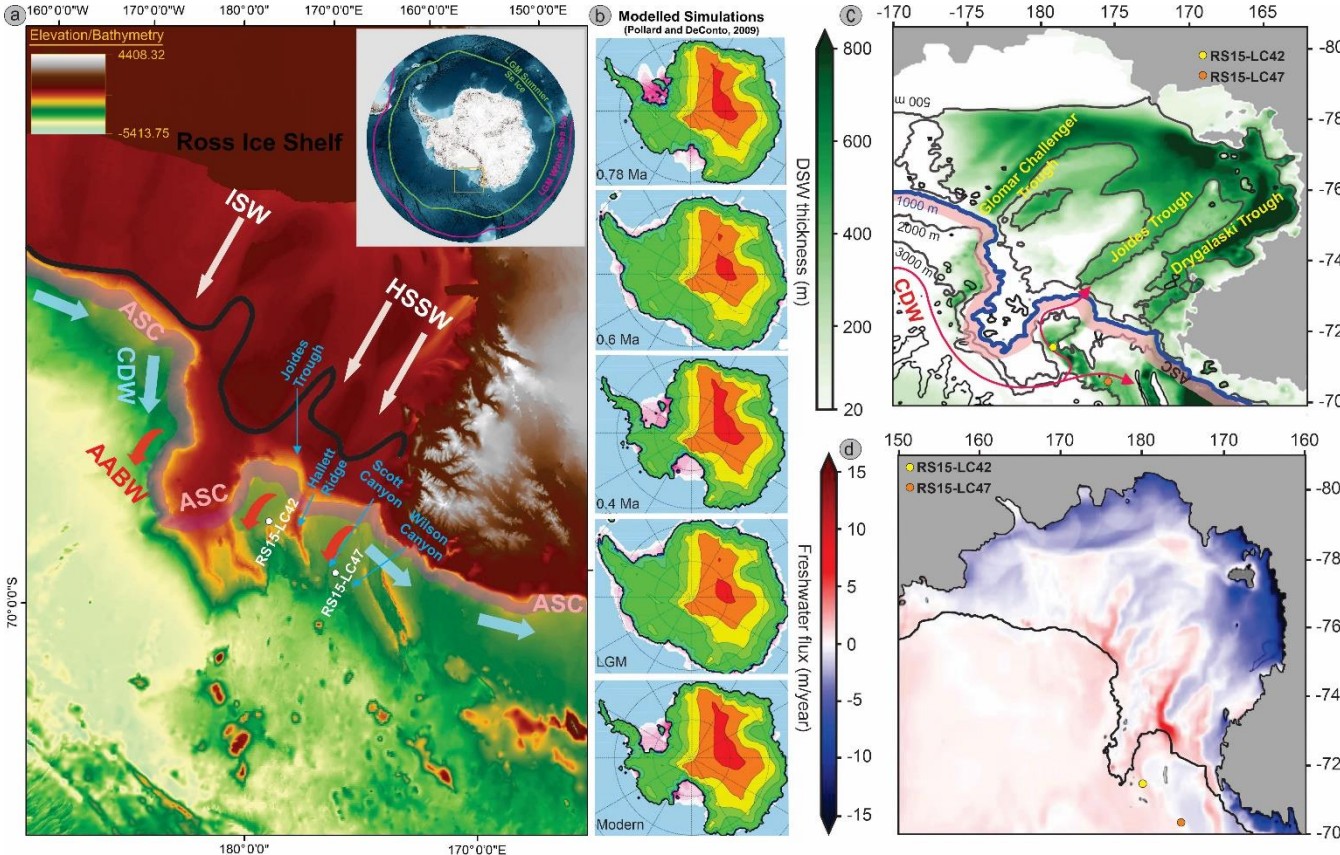

**Figure 1: (a) Regional map of the Ross Sea showing location of RS15-LC47 (this study) and RS15-LC42** (Bollen et al., 2022) **sediment cores. Bathymetry and topography are based on the ETOPO1 global relief model (https://www.ngdc.noaa.gov/mgg/global/). The inset map of Antarctica shows LGM summer and winter sea ice cover** (Green et al., 2022)**. Antarctic slope current (ASC), Antarctic bottom water (AABW), Circumpolar Deep Water (CDW), High Salinity Shelf Water (HSSW), and Ice Shelf Water (ISW) are modified from Smith Jr et al. (2012). (b) Modelled Antarctic ice volume changes since MPT (Pollard and DeConto, 2009). (c) Dense**

**Shelf Water (DSW) thickness shown in green shades** (Morrison et al., 2020)**, showing that energetic bottom water export plumes are**





**active at the study location, and likely drive sedimentological variability within the core. (d) Simulated Ross Sea surface freshwater flux** (Morrison et al., 2020)**.**

## 2 Materials and Methods

The RS15-LC47 gravity core (70°50'45.33"S, 175° 4'11.34"E) (Fig. 1) was collected from the Ross Sea lower continental rise during the Austral Summer 2015 by Korea Polar Research Institute IBR/Aron cruise at a water depth of 2417 m. The core is located at about 148 km South of Cape Adre, from a channel levee complex bounded by Scott and Wilson Canyons. A brief account of the oceanographic conditions at the study area is given in Supplementary Material I. The core was opened in Korea Polar Research Institute (KOPRI), and the litholog was prepared by visual description of the KOPRI scientists. We utilize

whole-core Magnetic Susceptibility (MS), sediment biogenic components (TOC, $CaCO_3$, and Biogenic silica), sedimentological features, and the isotopic ratio of authigenic Beryllium ($^{10}Be/^{9}Be$)$_{reac}$ to untangle the paleoenvironmental changes and their impact on depositional settings. Details of the methods are given in Supplementary Material II. Briefly, the chronology of RS15-LC47 sediment core was established by correlating MS values with that of a Relative Paleointensity (RPI) dated nearby sediment core (RS15-LC42: Bollen et al., 2022) (Fig. 2). The physical parameters and organic geochemical

proxies were measured at KOPRI as described by Kim et al. (2018a, 2020). For accurate paleoproductivity rate estimation, the sediment biogenic components were multiplied by the Mass Accumulation Rate (MAR) (Fig.2c) (Dymond et al., 1992). The MAR was derived by multiplying Linear Sedimentation Rates (LSR) with dry bulk density. Multielement analysis was performed using ICPMS for Ba, Al, Ti, and U at the Korea Basic Science Institute (KBSI), Ochang, Korea.

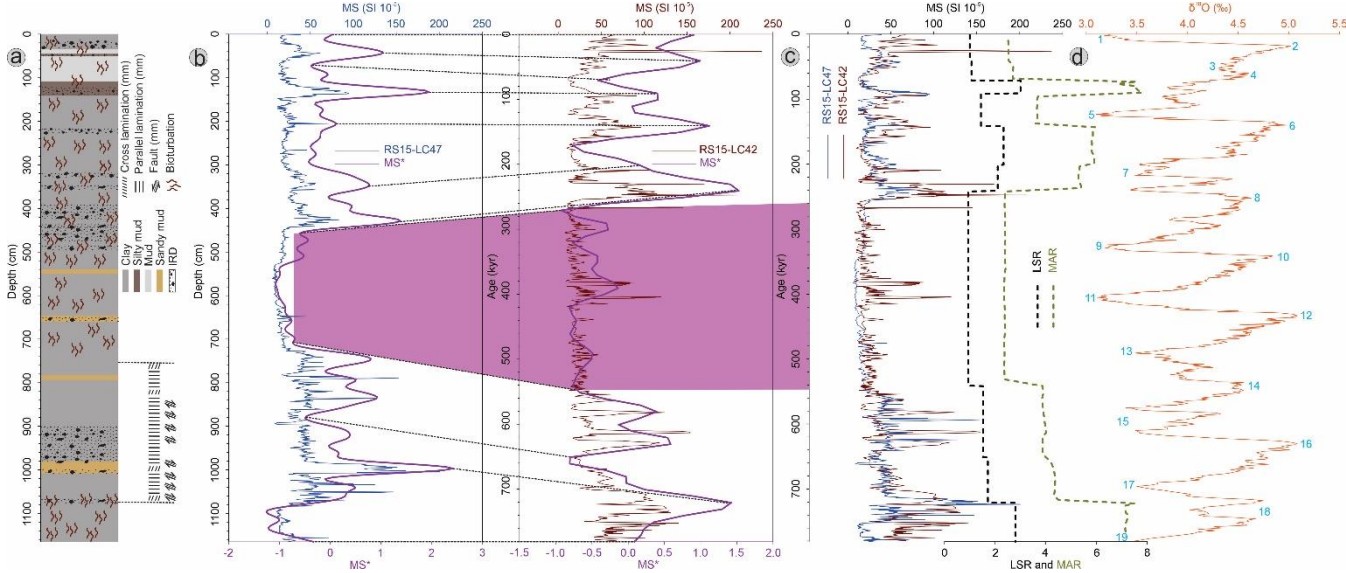

**Figure 2: (a) Lithology of the RS15-LC47 sediment core. Note: The lower segment of the studied interval (750-550 ka) displays distinct lithological characteristics than the rest of the core. It is characterized by parallel and cross laminations, along with**





**millimetre-scale faults; (b) Wiggle matching of MS values RS15-LC47 and RS15-LC42 sediment cores. The anomalously low MS values in both the cores were used as marker horizons (Purple box); (c) Age-integrated MS values of RS15-LC47 and RS15-LC42 sediment cores., linear sedimentation rate (LSR) and mass accumulation rate (MAR); (d) The stacked benthic foraminiferal δ18O**
**(LR-04 δ18O) (Lisiecki and Raymo, 2005).**

## 3 Results

### 3.1 Magnetic susceptibility, chronology, and lithostratigraphy

We employed a comprehensive approach by utilizing 12 tie points across both cores to establish a chronological framework,
and the correlation was effectively traced up to 781 ka (Fig. 2). The downcore variation in MS values in both cores (RS15-LC47 and RS15-LC42) reveals distinct patterns. The bottom part of the studied section exhibits low MS values between 781 and 750 ka. From 780-550 ka, high MS values are observed without a discernible trend, whereas in the middle sections (550-250 ka), the values dampen. Beyond 250 ka, the MS values exhibit a marked increase in variability. These observed patterns indicate a noteworthy similarity in both cores. The identified low MS values in the middle sections of both cores served as a
marker horizon for wiggle matching. Age-integrated MS values demonstrated a robust agreement between the two cores (Fig. 2c). The MS values for RS15-LC47 sediment core show a positive trend with the sand % and negative trends with silt % and clay % (Fig. 3), suggesting that coarser grains supplied during the glacial periods contributed to the enhancement of magnetic susceptibility. The stacked benthic foraminiferal $\delta^{18}O$ record does not exhibit a strong correlation with the MS (Fig. 2d). This lack of correlation may be attributed to the asynchronous nature of glaciation worldwide, as highlighted by Raymo et al. (2006).
Consequently, this $\delta^{18}O$ may not accurately capture specific ice volume changes in the Ross Sea region or the expansion/retreat of the West Antarctic Ice Sheet (WAIS).

From a lithostratigraphic perspective, the core sediment is predominantly characterized by clay and silt (Fig. 2a). The granulometric analysis also indicates that silt and clay are the prevailing grain sizes (Fig. 4I). Intermediate layers containing ice-rafted debris (IRD) and sandy mud are consistently present throughout the core. Notably, the IRD-enriched layers are
observed during the late Pleistocene period (> 250 ka) (Fig. 2a). The entire core exhibits evidence of bioturbation, with the exception of the interval between 720 and 1040 cm, corresponding to the time frame of 500-700 ka. During this interval, distinctive sedimentary structures such as parallel laminations and cross-beddings are apparent, deviating from the otherwise homogenous and structureless nature of the rest of the core (Fig. 2a). Within the mentioned depth range, ranging from 720 to 1040 cm, millimetre-scale faults are observable. These faults are interpreted as syndepositional, as they displace thin laminas
and are confined to a specific depth range.





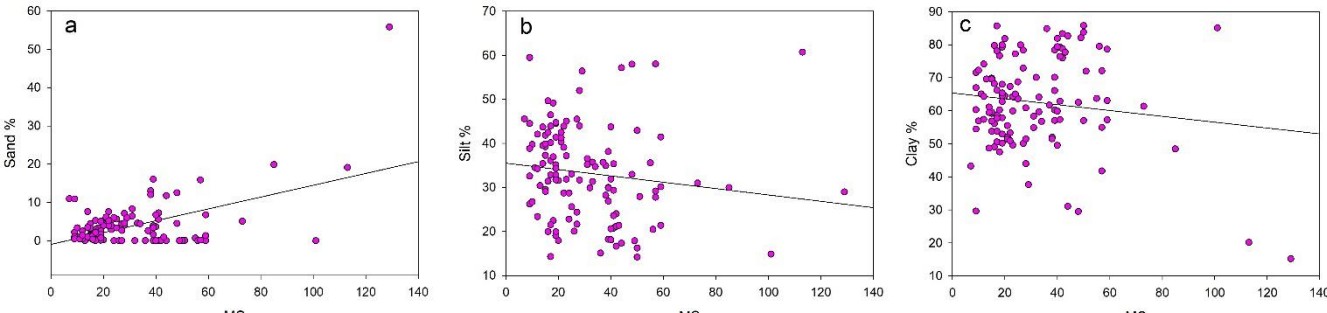

**Figure 3: Bivariate plots showing relation between different sedimentary components. (a), (b) and (c) show variation of MS values with respect to sand, silt and clay percentage, respectively.**

**3.2. Sedimentary organic geochemical proxies**

Figure 4 depicts the downcore variations in $CaCO_3$, TOC, C/N, Opal, and $Ba_{ex}$. The percentage variation of the biogenic components shows a similar trend with their MARs. $CaCO_3$, TOC, and C/N, exhibit markedly higher values between 550-750 ka. The covariation of both the percentage and MAR of these biogenic components suggests that the abnormally high values in this section are not affected by the sedimentation rate multiplier. Within this interval, $CaCO_3$ varies from 0 to 2.26 %

(Average = 0.43 %), the TOC varies from 0.06 to 0.41 % (Average = 0.25%), and C/N varies from 5 to 18 (Average = 10). Although, these organic geochemical parameters show remarkably lower values after 550 ka, still significant fluctuations are observed (Fig. 4II). Between 550-250 ka, the TOC and C/N vary from 0.07 to 0.24 (Average = 0.15) and 3.67 to 6.48 (Average = 5.04), respectively. After 250 ka, both TOC % (Average = 0.13 %) and C/N (Average = 5) significantly reduce and show a gradual increasing trend towards the top of the core. Visual correlation shows that $CaCO_3$ % shows a negative trend with TOC %

and C/N after 550 ka and has an average value of 0.025 %. Opal % does not show much variation till 250ka, however, it slightly increases and shows notable variation after that. After 250 ka the opal % varies from 9 to 22 %, with an average value of 16 %. $Ba_{ex}$, a productivity proxy in the Southern Ocean, was analysed from the 250-781 ka period and showed significant variations (Fig. 4I). $Ba_{ex}$ is significantly lower in the 750 to 550 ka interval as compared to the rest of the period. Interestingly, although TOC is significantly higher between 750 to 550 ka, other productivity proxies like $Ba_{ex}$ and Opal are lower in this

interval.



**Figure 4: Downcore variations of the different sedimentary components analysed in this study. (I) (a) Magnetic susceptibility (MS); (b) Sedimentation rate (SR) (cm/ka); (c) Dry Bulk Density (DBD) (g/cm³); (d) Mass Accumulation Rate (MAR) (g/cm²/ka); (e) Sand %; (f) Silt %; (g) clay %; (h) Sorting; (i) (¹⁰Be/⁹Be)ᵣₑₐc; (j) ¹⁰Beᵣₑₐc; (k) ⁹Beᵣₑₐc; (l) CaCO₃ %; (m) CaCO₃MAR; (n) TOC %; (o) TOCMAR; (p) C/N; (q) Opal %; (r) OpalMAR; (s) Ucrr (Detrital corrected U); (t) Baₑₓ (Barium excess); (u) Stacked benthic foraminiferal δ18O (LR-04 δ¹⁸O) (Lisiecki and Raymo, 2005). The broad climatic zones are represented with coloured boxes. The Purple box indicates relatively warmer phase, and the light blue coloured box indicates relatively colder phase. (II) The presence of abnormally elevated organic geochemical components within the 550-750 ka interval renders these parameters indiscernible for the subsequent period. The lower panel illustrates the fluctuation of organic geochemical components from 550 ka to the Present timeframe. The figure shows downcore variations of (a) TOC %; (b) TOCMAR; (c) CaCO₃ %; (d) CaCO₃MAR; (e) C/N; (f) Opal %; (g) OpalMAR; (h) (¹⁰Be/⁹Be)ᵣₑₐc; (i) Magnetic susceptibility (MS); (j) Stacked benthic foraminiferal δ18O (LR-04 δ¹⁸O; Lisiecki and Raymo (2005)) for 550 to Present timeframe.**



### 3.3. Variation of $^{10}Be_{reac}$, $^{9}Be_{reac}$ and $(^{10}Be/^{9}Be)_{reac}$

The concentration of $^{10}Be_{reac}$ (in atoms/g) exhibits a range from 3.79E+09 ± 1.15E+07 to 5.56E+07 ± 6.65E+05, with an average value of 1.48E+09 ± 5.58E+06. On the other hand, authigenic $^{9}Be_{reac}$ varies from 1.18E+17 ± 1.6E+16 to 2.03E+16 ± 3.82E+14, with an average of 6.76E+16 ± 1.85E+15. The $^{10}Be/^{9}Be$ ratio (in atoms/g) ranges from 6.26E-8 ± 1.29E-9 to 1.77E-9 ± 2.47E-11, with an average value of 2.16E-8 ± 5.96E-10 (Fig. 4). The downcore variations of both $(^{10}Be/^{9}Be)_{reac}$ and $^{10}Be_{reac}$ follow a similar pattern. They exhibit an increasing trend towards the late Pleistocene, with the lowest values observed in the

550-781 ka interval. Interestingly, the trends of $(^{10}Be/^{9}Be)_{reac}$ and $^{10}Be_{reac}$ show opposite trend with MS values but align with Opal % and TOC % after 250 ka.

### 4 Discussion

### 4.1. Sediment facies association and depositional environment

Magnetic susceptibility, along with sediment lithological characterization, have been used as primary tools to interpret paleoenvironment from high latitude sediment cores (Mead et al., 1986; Diekmann et al., 2000; Pirrung et al., 2002; Brachfeld, 2006; Bollen et al., 2022). We interpret the sediment depositional environments based on visual observations and physical properties. In addition, we also take advantage of the sedimentological parameters of RS15-LC47 sediment core reported by Al'bot (2016) (unpublished) for the interpretation.


### 4.1.1. Late-MPT ice sheet-proximal depositional environment

The lower segment of the studied interval (750-550 ka) displays distinct lithological characteristics than the rest of the core. It is characterized by parallel and cross laminations, along with millimetre-scale faults. The presence of laminated clay-rich layers, infrequent sandy layers, the absence of bioturbation, and low sorting in this segment suggest an origin associated with

contourite processes (McCave et al., 1980; Faugères et al., 1993; Howe et al., 1997; Koenitz et al., 2008; Faugères and Mulder, 2011; Thiéblemont et al., 2019). The well-defined and continuous laminations result from consistent bottom currents and are well-preserved due to the lack of bioturbation (Weber et al., 1994; Rodrigues et al., 2022). The presence of IRD layers within the laminated mud eliminates the possibility of a turbiditic origin (Rodrigues et al., 2022). However, it's noteworthy that low-density turbidity currents can still transport ice-rafted debris to deeper oceans (Wang and Hesse, 1996). The occurrence of

cross-laminated mud in this interval can be attributed to the intermittent sedimentation by turbidity currents (Lucchi et al., 2002; Rodrigues et al., 2022).





This temporal span is within the Mid-Pleistocene Transition (MPT) (1200-600 ka), characterized by the expansion of the Antarctic ice sheet (McClymont et al., 2013; Sutter et al., 2019) and northward migration of the polar front (Kemp et al., 2010). During periods of full glaciation, the ice sheet extends to the outer continental shelf, restricting the Antarctic Slope Current (ASC) to a narrower slope (Conte et al., 2021). This confinement limits energy dissipation and water exchanges, potentially intensifying the ASC, which plays a crucial role in regulating Circumpolar Deep Water (CDW) inflow and Ross Sea Bottom Water (RSBW) outflow. During full glacial intervals, the expansion of the ice sheet across the Antarctic continental shelf is evident from the erosional surfaces in the sedimentary strata (McKay et al., 2012b; Bollen et al., 2022). In the Ross Sea sector, there have been multiple instances of the grounding line advancing toward the shelf break during glaciations since the Mid-Pleistocene Transition (MPT) (McKay et al., 2012b). This advance of the grounding line triggers sediment gravity flows and turbid plumes and channelizes them to the outer continental margin through the canyon networks (Fig. 5c) (Larter and Barker, 1989; Lucchi et al., 2002). Thus, the mm scale faults observed in this interval can be categorized as synsedimentary, which developed due to higher sedimentation by turbid plumes. The fine fraction of these flows settles gradually and is transported westward by along-slope currents. Downslope sediment gravity flows, and along-slope currents play pivotal roles in shaping channel-levee systems around the Antarctic margin during glacial periods. The higher clay percentage and MAR, and the low sorting observed at the studied site during this time interval align with this sedimentation pattern (Fig. 4). The higher sedimentation rate and hostile glacial conditions diminished the biological activity during this period. Higher values and irregular trend in MS during this interval suggests poor sorting, supporting sedimentation dominated by gravity flows. McKay et al. (2012a) interpreted sediment winnowing using MS record. Enhanced sediment winnowing occurred during colder intervals due to the strengthening of the Antarctic Slope Current (ASC).

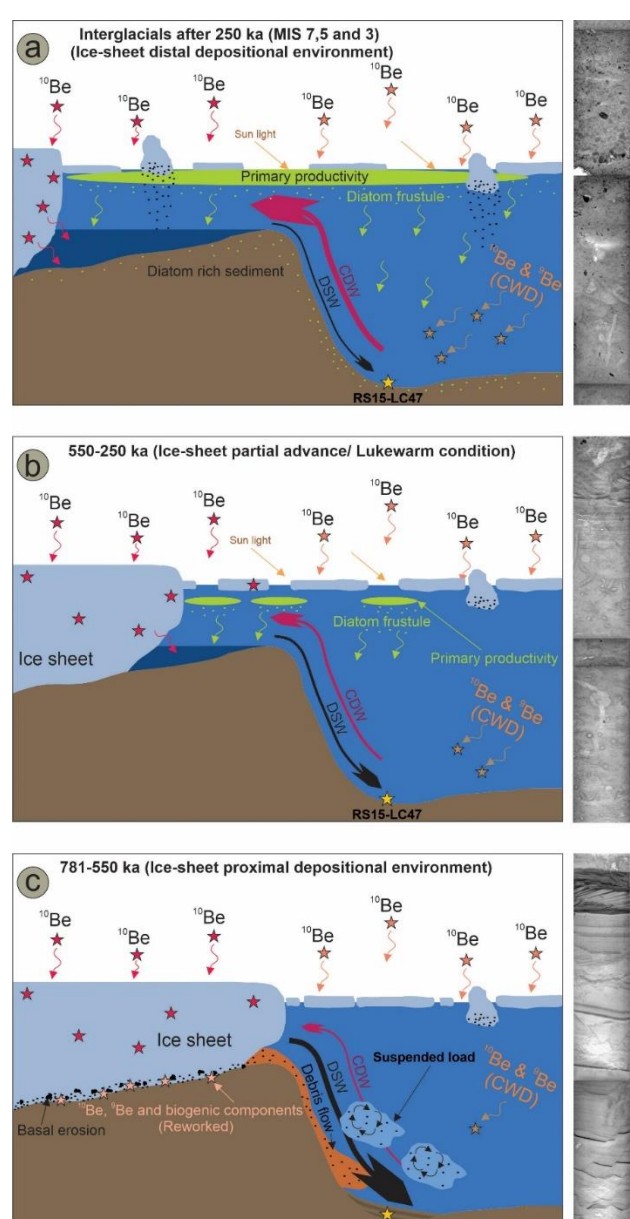

**Figure 5: Illustrates hypothesized oceanographic conditions in the Ross Sea spanning the past 781 ka, accompanied by representative X-ray images for the depicted time frames. (a) Ice-sheet distal depositional setting during the interglacials of 250 ka to the present timeframe, characterized by strengthened Circumpolar Deep Water (CDW) flow, resulting in high ($^{10}$Be/$^9$Be)$_{reac}$. Enhanced surface productivity and $^{10}$Be flux are observed due to expansive open sea conditions. DSW production declines due to reduced ice coverage; (b) Lukewarm condition during 550-250 ka with partial advance of sea ice. CDW influx remains moderately high, yielding relatively elevated ($^{10}$Be/$^9$Be)$_{reac}$ compared to glacial periods. Surface productivity and $^{10}$Be flux also rise owing to favourable open sea conditions; (c) Ice-sheet proximal depositional environment during 781-550 ka. Intensified DSW production triggers gravity flows, mobilizing shelf sediments that settle on the outer continental margin. Reduced CDW influx, increased sea ice coverage, strengthened DSW supply, and mobilization of 10Be-depleted older depositional units collectively contribute to low $^{10}$Be/$^9$Be)$_{reac}$ levels during this period.**





### 4.1.2. Partial-advancement of ice-sheet during mid-late Pleistocene

Following the Mid-Pleistocene Transition (MPT), MS values remained consistently low until MIS 8 (~ 250 ka) in both the
cores (RS15-LC47 and RS15-LC42), suggesting persistent interglacial during this timeframe and probably ice sheets attained
a stable configuration. Gravel counts are low (Al'bot, 2016), and IRD-rich layers are sparse in this interval. Clast-rich sediment
layers at the Antarctic marginal settings are attributed to the action of bottom currents, where winnowing of finer fractions due
to robust bottom currents enhance the abundance of gravelly clasts (Drewry and Cooper, 1981; Bollen et al., 2022). A notable
absence of clast-rich sediment layers during this timeframe suggests diminished bottom current action. Ice sheet models divide
the post-MPT period into two episodes based on ice sheet extent (Sutter et al., 2019). The first part, spanning till MIS 11, is
characterized by smaller glacial ice sheets and prolonged lukewarm interglacials. We speculate subdued glacials during this
timeframe due to the influence of prolonged lukewarm condition. Hesse et al. (1997, 1999) suggest that sedimentation from
meltwater plumes is limited to a few tens of kilometres near the coast during interglacial periods. The wider landward sloping
continental shelf off the Pacific margin of the Antarctic Peninsula acts as a sediment trap during interglacials, hindering the
supply of coarser sediments to the continental rise. The low percentages of sand and gravel (Fig. 4), along with a notable
absence of IRD-rich layers in this interval, suggest that neither of the processes supplying coarser sediments to the outer
continental margin, i.e., ice-sheet calving during interglacials or mobilization of sub-ice shelf sediments during glacials, were
dominant during this time. The ice sheets probably remained relatively stable near the coastline, preventing the movement of
coarser sub-glacial sediments or those resulting from ice-sheet calving to the offshore region (Fig. 5b).
Despite these overall trends, a closer look at granulometric parameters reveals two distinct sedimentation episodes within this
interval: from MIS 15 to 11 and from MIS 11 to 8 (Fig. 4I). Although the sand % remains relatively stable, silt and clay
percentages, along with sorting, exhibit differing trends between the two episodes. The former is characterized by well-sorted
sediments with a lower clay and higher silt percentage compared to the latter. While the clay percentage differs by a modest
10% between the two phases, there is a more substantial (~ 30%) variation in silt percentage, along with noticeable differences
in sorting. This implies that even though the gravel-sized clasts from the freshwater plumes were trapped on the continental
shelf when the ice sheets were proximal to the coastline, the interglacial meltwaters could still transport silt-sized sediments
farther offshore. Thus, based on the granulometric analysis, we interpret two distinct and stable environmental conditions
during this interval, the former being relatively warmer than the latter. Sediment facies association from the AND-B sediment
core also supports similar sedimentation environments during the mid-late Pleistocene, with grounding-line proximal deposits
during the former episode and distal deposits during the latter (McKay et al., 2012b).

### 4.1.3. Warmer late Pleistocene

The late Pleistocene period (> 250 ka) is characterized by coarser grain size, IRD-rich layers, and high-frequency variations
in MS values (Figs. 2 and 4 I). The MS values are low during the interglacials of MIS 7, 5 and 3, suggesting sedimentation





distal to ice-sheet (Fig. 5a). As reported by Al'bot (2016), the gravel counts also gradually increase after MPT, which is
significant after 250 ka. The general sedimentation pattern in the Antarctic continental margin deposits indicates clast-dominated sediments during the late Pleistocene (Bollen et al., 2022). Ice sheet models and proxy records from the Antarctic continental margin suggests that after MIS 11 the interglacial temperature raised by 1-2° and ice volume variability increased, with smaller interglacial and larger glacial ice sheets (Sutter et al., 2019). Although it is challenging to differentiate specific episodes of gravel/IRD enrichment, the presence of clast-rich layers and absence of laminations suggests intensified bottom
current and sediment deposition in high-energy conditions.

### 4.2. Pleistocene ice sheet dynamics inferred from Beryllium isotopes

### 4.2.1. $^{10}$Be inventory to the Ross Sea

The Be isotopic record from this study is as par with that of open marine settings in the Ross Sea region (in the order of $10^9$) (Sjunneskog et al., 2007; Yokoyama et al., 2016) and higher than that from the sub-ice shelf deposits, suggesting the influence
of local environmental settings on the Be isotope inventory. The measured $^{10}$Be in the Ross Sea region is two orders of magnitude higher than the hypothetical concentration during Holocene (in the order of $10^7$, assuming an atmospheric flux rate of $5\times10^5$ atoms/cm$^2$) (Valletta et al., 2018). This indicates that atmospheric deposition alone cannot explain the observed variations of $^{10}$Be concentrations in glacio-marine sediments, emphasising the need to understand additional sources to elucidate the observed concentrations. The variability in $^{10}$Be/$^9$Be at the Antarctic continental margin is proposed to have been
contributed by pulsed $^{10}$Be delivery by basal melt (Valletta et al., 2018). An increasing trend observed in $^{10}$Be concentration during the interglacials is attributed to increased basal melt and higher scavenging rate. Nevertheless, it is acknowledged by previous researchers that atmospheric flux alone is inadequate to explain the observed $^{10}$Be inventories along the Antarctic coastline during interglacials (Sjunneskog et al., 2007; Valletta et al., 2018).

The melt water and atmospheric flux sourced $^{10}$Be inventory and deposition in the Ross Sea ($^{10}Be_{RS}$) was calculated using:

$$^{10}Be_{RS} = \frac{T_{melt}\, \rho_{ice}\, ^{10}Be_{ice} + \, ^{10}Be_{flux}\, A_b}{A_b SR\, \rho_{sed}}$$ (Behrens et al., 2019)

The total meltwater ($T_{melt}$), annual volume of meltwater delivered by the grounding line, basal melt, and iceberg calving, from eleven major glaciers – Land, Nickerson, Sulzberger, Swinburne, Withrow, Ross West, Ross East, Drygalski, Nansen, Aviator, and Mariner – flowing into the Ross Sea was quantified (Rignot et al., 2013). Our calculations incorporated a sediment density of 2.5 g/cm³ (average value in this study) and an ice density of 0.9 g/cm³ (Behrens et al., 2019). We assumed a uniform
$^{10}Be$ concentration in glacial ice ($^{10}Be_{ice}$) draining into the Ross Sea Embayment, with a maximum average value of $2 \times 10^5$ atoms/cm³, accounting geomagnetically induced $^{10}$Be-flux variability during the Pleistocene (Sjunneskog et al., 2007). The basin area of the Ross Sea ($A_b$) was determined to be $4.9\times10^3$ km$^2$ (Comiso et al., 2011). Additionally, we utilized an average atmospheric $^{10}$Be flux value of $4.5 \times 10^5$ atoms/cm² for the Ross Sea (Lal, 1987). $^{210}$Pb derived average maximum sedimentation rate of 0.07 cm/yr for Ross Sea region was used for calculation (Bettoli et al., 1998). The concentration of $^{10}$Be
in sediments from the Ross Sea was calculated to be $8.9\times10^7$ atoms/g, a stark deviation of two orders of magnitude from the





estimated value of approximately $1.5 \times 10^9$ atoms/g by Sjunneskog et al. (2007), which was based on a presumed basal melt rate of 30 cm. Similarly, this value differs significantly, also by two orders of magnitude, from both the measured values in our study and the reported values in open marine settings of the Ross Sea (Sjunneskog et al., 2007). Although underestimation of melt water contribution or overestimation of the sedimentation rate may bias this calculation, a significant factor that needs

to be considered in the $^{10}Be_{reac}$ inventory estimations is the Circumpolar Deep Water (CDW) influx (Jeromson et al., 2024). If the increase in meltwater flux were the primary driver behind the elevation of the $(^{10}Be/^9Be)_{reac}$ ratio in sediment records, one would anticipate the highest $(^{10}Be/^9Be)_{reac}$ ratios occurring where meltwater is most abundant. However, intriguingly, Antarctic marginal seas exhibit low $(^{10}Be/^9Be)_{reac}$ signatures where meltwater is most prevalent, particularly at ice shelf outflow sites compared to other open water locations (Sjunneskog et al., 2007; Jeromson et al., 2024). Sjunneskog et al. (2007) noted a

significant difference in $^{10}Be_{reac}$ levels between the open water regions and the continent-proximal settings of the Ross Sea, with the former showing a two-order magnitude increase. Jeromson et al. (2024) expanded on this, indicating that while about 20% of $^{10}Be$ is contributed from meltwater flux, the primary source, constituting over 60%, is the upwelling of Circumpolar Deep Water (CDW). This spatial pattern consistently shifts from higher values nearer to the point of CDW upwelling to lower values in closer proximity to terrestrial sources. Given the predominant supply of $^{10}Be_{diss}$ from CDW, and the anticipated high

$(^{10}Be/^9Be)_{reac}$ ratios within CDW due to its deep-water origin, transitions from conditions with absent or reduced upwelling to those with present or intensified upwelling throughout the Quaternary are expected to correlate with a transition from low to high $(^{10}Be/^9Be)_{reac}$ ratios.

### 4.2.2. Relative variation of $^{10}Be_{reac}$ and $^9Be_{reac}$

While individual beryllium isotopes $^{10}Be_{reac}$ and $^9Be_{reac}$ may not independently offer reliable insights into paleoenvironmental

conditions, their ratio $(^{10}Be/^9Be)_{reac}$ proves valuable (Valletta et al., 2018; Behrens et al., 2022). The decoupling between $^{10}Be_{reac}$ and $^9Be_{reac}$ becomes evident in their normalized plots, and a homogeneity would result in a 1:1 relationship. The observed trendline in the $^{10}Be$ vs $^9Be$ plot for the RS15-LC47 sediment core exhibits an overall slope less than 1, indicating an additional input of $^{10}Be$ (Fig. 6). However, the trendline exhibits variability across different sedimentation environments, as discussed in Section 4.1 (Fig. 6). During warmer intervals, the slope of the trendline is <1, while during colder episodes, it is

>1. This variability in the $^{10}Be_{reac}$ vs $^9Be_{reac}$ relationship suggests an increased flux of CDW sourced $^{10}Be_{reac}$ during the warmer interglacial periods. The $^9Be_{reac}$ isotope is interpreted to primarily originate from glacial meltwater and basal erosion, is delivered to the deep ocean via Antarctic Bottom Water (ABW) (Valletta et al., 2018).

The average $^{10}Be$ concentration in the studied core ($1.48 \times 10^9$ atoms $g^{-1}$) aligns with that found in open marine hemipelagic settings of the Southern Ocean ($1.5 \times 10^9$ atoms $g^{-1}$, Sjunneskog et al., 2007). However, it notably exceeds the concentration

in the nearby sediment core RS15-LC42 by an order (Dash et al., 2021). Despite both cores being retrieved from the continental rise region, this discrepancy in $^{10}Be$ concentration can be attributed to local geographical settings (Sjunneskog et al., 2007). The RS15-LC42 sediment core, obtained from the downslope terminus of the Joides Trough (Fig. 1), is exposed to dense slope water (DSW) plumes and gravity flows. Consequently, this site is subject to basal meltwater flux and deposition of glacially



reworked older sediments. During glacial maxima, the Ross Sea continental shelf becomes ice-covered, channelling scoured

sediments through narrow slopes (Fig. 5a), for RS15-LC42 it is Joides Trough. We speculate that the observed low $^{10}$Be

concentration results from dilution with $^{10}$Be-depleted basal meltwater and older sediments. This older sediment dilution

hypothesis is supported by the presence of reworked diatoms between 800 ka to 300 ka in the RS15-LC42 sediment core

(Bollen et al., 2022), predating the onset of the post-Mid-Bruhnes Event (MBE) warmer conditions. A similar explanation may

apply to relatively lower $^{10}$Be concentrations in settings similar to RS15-LC42, such as the downslope of the Adélie Depression

(Valletta et al., 2018), Ross Sea sediment cores from past ice stream flow paths (Sjunneskog et al., 2007), and the Adélie Basin

(Behrens et al., 2019).

**Figure 6**: **Illustrates a normalized plot of $^{10}$Be versus $^9$Be for various timeframes as elaborated in Sections 4.1 and 4.2. A slope greater than 1 signifies enhanced $^9$Be delivery resulting from basal erosion during ice-sheet advancement. Conversely, a slope less than 1**

**indicates augmented $^{10}$Be supply from additional sources (CDW).**

### 4.2.3. Paleoenvironmental/ paleoceanographic signal from ($^{10}$Be/$^9$Be)reac



The $(^{10}Be/^9Be)_{reac}$ (Be isotope ratio) is lowest in the interval 550-781 ka. This interval belongs to the later part of the MPT, characterized by extended glacial conditions (Fig. 5c) (Clark et al., 2006; Bol'Shakov, 2015; Daruka and Ditlevsen, 2016;
Herbert, 2023). This period signifies a sudden reorganization of the climate system, occurring in at least two discernible stages, i.e., MIS 25-22 and MIS 22-16. From MIS 25 to 22, there was a noteworthy augmentation in ice volume in both the West and East Antarctic Ice Sheets, contributing to the commencement of a full glacial period around MIS 22 (approximately 900 ka) (Ford et al., 2016). The MIS 16 is conferred as the period with maximum global ice volume. Throughout the MIS 22 to MIS 16 timeframe, the climate system experienced modest interglacial conditions interspersed with progressively more severe
glacial phases. During this period, the ASC strengthened reducing CDW inflow. The low $(^{10}Be/^9Be)_{reac}$ isotopic ratio resulted from weakened CDW influx and low $^{10}Be_{reac}$ input to the water column due to enhanced sea ice coverage (Fig. 5c).

The $^{10}Be_{reac}$ and the $(^{10}Be/^9Be)_{reac}$ values show a notable increase following MIS 16, particularly during MIS 14-11, indicating an extended interglacial duration. This observation aligns with the evidence of enhanced sorting and higher silt percentage during this interval. In contrast to mid-late MPT period, a subsequent south-westward shift of the ASL and dominance of
easterlies after MIS 16 redirected the sea ice regime toward the Amundsen Sea (Hillenbrand et al., 2009). This altered scenario, with a southward shift of ASL and an increased advection of Circumpolar deep water towards the shelf zone could have caused instability in the Antarctic ice sheets. While the $(^{10}Be/^9Be)_{reac}$ values are higher compared to the preceding interval, they remain lower than the late Pleistocene (> 250 ka) values observed in our study. We characterize this interval as a "lukewarm" phase due to its extended duration and moderately high $(^{10}Be/^9Be)_{reac}$ ratio (Fig. 5b). Previous studies on Antarctic marginal seas
have also documented an extended interglacial period following the MIS 16, which researchers suggest may have contributed to the potential collapse of the WAIS. Hillenbrand et al. (2009) suggest the possibility of WAIS collapse occurring between MIS 15-13, drawing from analyses of productivity and lithogenic proxies in the Amundsen Sea. They propose that changes in circulation patterns subsequent to MIS 16 may have triggered this collapse, noting anomalous increases in biological productivity and lithogenic supply during the period spanning 621–478 ka. Similarly, Scherer et al. (1998) report a potential
WAIS collapse during MIS 11, based on observations of increased $^{10}Be$ concentration in sediments beneath the WAIS. They attribute this increase to open marine conditions following the disintegration of the ice sheet. In support of this, Hearty et al. (1999), Hearty et al. (1999), and Kindler and Hearty (2000) present findings indicating notable deglaciation events between 570 and 390 ka. The discrepancy in proposed timing for the collapse of the WAIS may stem from chronological constraints. However, the period from MIS 16 to MIS 11 remains the most contentious timeframe for such a collapse. Although the
observed $^{10}Be_{reac}$ and $(^{10}Be/^9Be)_{reac}$ increase during the MIS-14-11 in our study does not provide a direct evidence for WAIS collapse, it indicates a prolonged lukewarm condition favourable for the ice-shelf disintegration.

The $(^{10}Be/^9Be)_{reac}$ increases significantly after 250 ka, showing opposite trend to the MS values (Figs. 4&7a). It is proposed that interglacial temperatures have risen by 1–2 °C since 300 ka. The incursion of warm water onto the continental shelf during the interglacials has been extensively documented across various regions of the Antarctic margin, spanning from the Ross Sea
and Amundsen Sea to the Antarctic Peninsula (Howe and Pudsey, 1999; Steig et al., 2012; King et al., 2022; Yi et al., 2023; Lamy et al., 2024). This influx of warm Circumpolar Deep Water (CDW) directly influences glacial dynamics by fuelling



escalated melt rates, consequently triggering retreat and collapse of ice shelves. The intensified intrusion of CDW during the interglacials can be attributed to two main factors: firstly, diminished formation of dense water on the shelf in the Ross Sea, and secondly, a weakened Antarctic Slope Current (ASC), acting as a hydrographic barrier that typically restrains CDW intrusion. This upwelling of CDW facilitated increase in the $(^{10}Be/^{9}Be)_{reac}$ ratio during this period. Notably, there exists an inverse correlation between magnetic susceptibility (MS) values and the $(^{10}Be/^{9}Be)_{reac}$ ratio, which supports the hypothesis that MS values tend to peak during glacial periods (Fig. 7a).

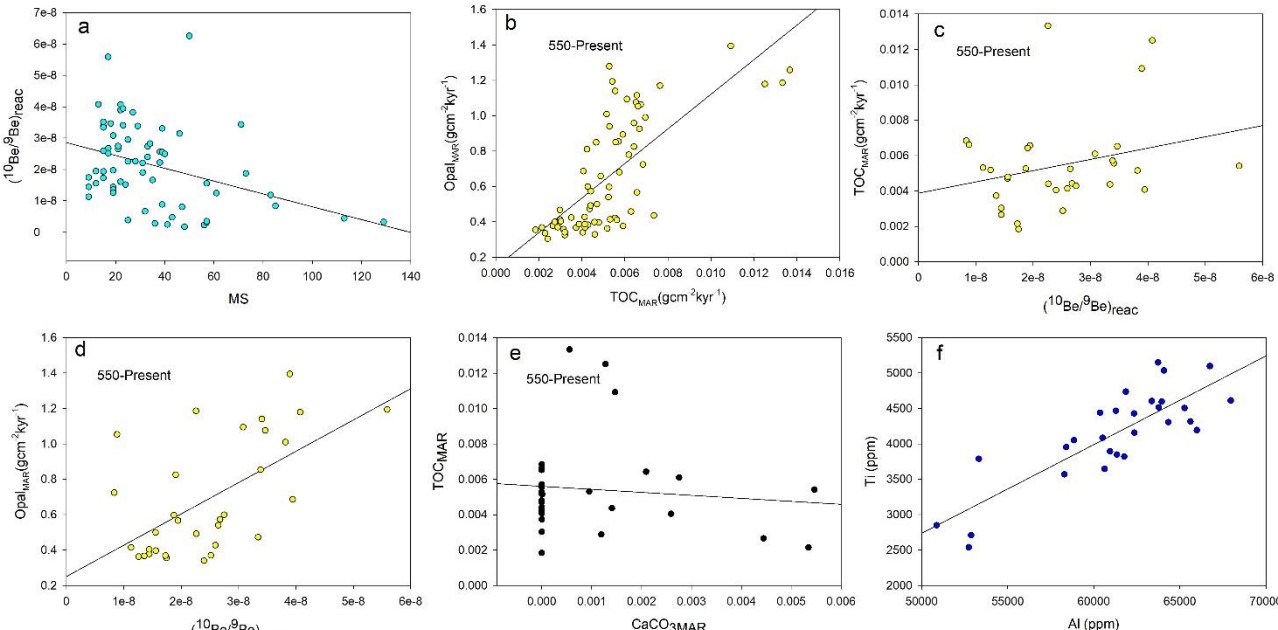

**Figure 7: Bivariate plots between (a) $(^{10}Be/^{9}Be)_{reac}$ vs MS; (b) Opal$_{MAR}$ vs TOC$_{MAR}$; (c) TOC$_{MAR}$ vs $(^{10}Be/^{9}Be)_{reac}$; (d) Opal$_{MAR}$ vs $(^{10}Be/^{9}Be)_{reac}$; (e) TOC$_{MAR}$ vs CaCO3$_{MAR}$; (f) Ti vs Al. Note: The bivariate plots depicting sedimentary organic geochemical components have been limited to 550 ka due to the anomalous increase in their concentrations beyond this point, interpreted to originate from reworked shelf sediments.**

## 4.3. Production and preservation of biogenic material at the Ross Sea continental margin

### 4.3.1. Opal in Drift Sediments of the Antarctica: Relevance to Biological Productivity or Preservation Dynamics?

The depositional environments of sedimentary organic geochemical components in this study is interpreted based on the trends of downcore variations (Fig. 4) and the interparametric relationships (Fig. 7). Despite the various dissolution processes, opal records in upper Quaternary sediments from Antarctica are interpreted as proxies for climate-induced changes in surface water productivity (Charles et al., 1991; Hillenbrand and Fütterer, 2001; Grützner et al., 2005; Hillenbrand and Ehrmann, 2005; Weber et al., 2022). However, discrepancies between opal, total organic carbon (TOC), calcium carbonate (CaCO3), and





biogenic barium (Ba$_{ex}$) records are observed for the 781-550 ka interval (Fig. 4I). Opal content shows insignificant variation until 250 ka, increasing significantly afterward. Discrepancies also exist between opal content and its accumulation rates (Opal$_{MAR}$) during this interval. Despite this, visual observation shows that downcore variation of both opal content and Opal$_{MAR}$ exhibit a positive trend with ($^{10}$Be/$^9$Be)$_{reac}$ and a negative trend with magnetic susceptibility (MS) after 250 ka (Figs. 4I&II),

indicating increased surface productivity during warmer climatic intervals of late Pleistocene and vice versa.

In general, Opal$_{MAR}$ is influenced by opal deposition and preservation, while opal content may be additionally affected by dilution with other sedimentary components, primarily lithogenic particles drift sediments. During the late Pleistocene at our study site, no significant discrepancies are observed between opal contents and Opal$_{MAR}$, but differences are significant during the Mid-Pleistocene Transition (MPT), coinciding with notably high linear sedimentation rates (LSR). This suggests that opal

content records in drift sediments were significantly masked by lithogenous particle dilution. The original depositional rates of siliceous particles are strongly influenced by changes in LSR, with lateral redistribution of biosiliceous particles by bottom currents further affecting Opal$_{MAR}$. Bottom currents likely acted as focusing mechanisms on the Antarctic Peninsula continental rise, enhancing the delivery of interglacial productivity signals from overlying water columns. Conversely, turbidity currents from unstable locations at the continental slope mainly resuspend lithogenic components, explaining the low opal contents in

MPT sediments. Nonetheless, extensive occurrences of reworked diatoms along the Ross Sea continental margin during the MPT (Bollen et al., 2022) suggest lateral redistribution of biosiliceous matter from the shelf to the continental rise, contributing to Opal$_{MAR}$  computations for the 781-550 ka period.

During the period spanning 781 to 550 ka, there is a noticeable spike in Total Organic Carbon (TOC) and Carbon-to-Nitrogen (C/N) ratio (Fig. 4I). The average C/N value in this timeframe reaches 11, significantly higher than the Redfield ratio of 6.6 and exceeding the natural variability observed in the euphotic layer of both the Ross Sea (5.4 – 9.2; Fabiano et al., 1993) and

the Amundsen Sea polynya (6.3 – 9.2; Kim et al., 2018a). The C/N ratio serves as an indicator of sediment source, with higher values suggesting a dominance of terrestrial organic carbon (Sarmiento, 2006; Li et al., 2016). The increase in TOC/TN reflects a peak in terrestrial input likely due to advancing ice sheets, which could have led to reduced marine productivity and an increase in glaciomarine influence. Additionally, the decrease in the Al normalised Barium (Ba$_{ex}$), a proxy for productivity,

during this period suggests diminished productivity (Fig. 4I). Although Al in oceanic sediments may also be contributed from biogenic sources, other than terrigenous input, the Al shows a high positive correlation with Ti, suggesting that the Al is primarily terrigenous sourced (Fig. 7f). Biogenic barium signals are considered resistant to dissolution processes in Antarctic continental rise settings due to prevailing oxic conditions in the sediment column (Shimmield et al., 1994; Bonn et al., 1998). The detrital corrected U record from our core also suggests that Ba$_{ex}$ is not affected by anoxic dissolution. Based on the

relationship between TOC, C/N and Ba$_{ex}$ it is hypothesized that the elevated organic carbon during this period might have originated from reworked bottom sediments during the glacial advance.

In contrast, the period between 550 and 250 ka shows TOC/TN levels within the range observed in the euphotic layer of the Ross Sea (Fig. 4II). Ba$_{ex}$ values are relatively higher during this interval, correlating with TOC levels. Therefore, the increased TOC values during this period are likely due to higher productivity in an extended lukewarm condition, supported by



anomalously lower magnetic susceptibility (MS) values and relatively higher $(^{10}Be/^9Be)_{reac}$ ratios compared to the preceding interval. Post 550 ka, Opal % shows a positive correlation with $(^{10}Be/^9Be)_{reac}$ and TOC (Fig. 7b-d). This suggests increased late Pleistocene productivity during warmer periods and vice versa. The C/N ratio exhibits a negative trend with TOC % and Opal % during this period (Fig. 4II). During interglacial periods, the decrease in C/N ratios suggests a correlation between elevated biogenic opal and TOC concentrations and increased surface water productivity. This relationship arises from the

characteristic lower C/N ratio of marine organic matter compared to terrestrial sources.

### 4.3.2. Variation in calcium carbonate and its significance for bottom water dynamics during the mid- late Pleistocene

Analysis of deep-sea cores in the Southern Ocean unveils diverse patterns in calcium carbonate preservation, likely influenced by fluctuations in bottom water alkalinity (Emerson and Archer, 1990; Rickaby et al., 2010; Steiner et al., 2021). For instance, cores retrieved from the Ross Sea exhibit preservation trends similar to those found in the Pacific, showing enhanced

preservation during glacial periods and deglaciations (Bonaccorsi et al., 2007). In contrast, cores from locations within the Antarctic Circumpolar Current demonstrate an opposite pattern reminiscent of the Atlantic.

In the RS15-LC47 sediment core, peaks in calcium carbonate content predominantly align with colder episodes, suggesting enhanced preservation during the glacials of the 781-550 ka interval. This observation hints at a reduction in water column corrosiveness in the Ross Sea during glacial phases. Such a phenomenon may arise from the buildup of siliceous detritus on

the seabed, potentially facilitating organic matter decay and the dissolution of calcareous foraminifers during interglacial periods (Kennett, 1968). Conversely, during glacials, robust bottom currents likely increase calcium carbonate preservation by curbing the deposition of fine materials (Taviani et al., 1993; Taviani and Claps, 1998; Frank et al., 2014). This notion gains further support from the observation that increases in calcium carbonate content coincide with intervals of low biogenic opal and total organic carbon concentrations (Fig. 7e), along with elevated magnetic susceptibility values (Fig. 4II).

## 5. Conclusions

This study employs a comprehensive approach to unravel the paleoenvironmental changes in the Ross Sea over the past 800 ka. In the studied timeframe, the 750-550 ka period exhibits anomalous depositional signature with presence of parallel and cross laminations, as well as mm-scale faults. Based on the sedimentological characteristics we interpret this depositional pattern linked to contourite process. This period belongs to the late Mid-Pleistocene Transition (MPT) (1200-600 ka), marked

by significant expansion of ice sheets. The low $(^{10}Be/^9Be)_{reac}$ isotopic ratio during this period suggests a weakened influx of Circumpolar Deep Water (CDW), possibly due to glacial strengthening of the Antarctic Slope Current (ASC). We speculate that grounding line extended to the shelf edge during this time frame and sedimentation occurred by energetic meltwater plumes, density-driven currents, and slope failures. The sediment organic geochemical components are also anomalously high in this segment. Although Total Organic Carbon (TOC) is high during this time period, higher Carbon-to-Nitrogen (C/N) ratios

along with relatively lower $Ba_{ex}$ concentrations does not suggest increase in productivity, rather hint at the reworking of bottom

sediments during glacial advances contributing to increased organic carbon levels. Following the Mid-Pleistocene Transition (MPT), MS values remained consistently low until MIS 8 in both the cores (RS15-LC47 and RS15-LC42). This period is interpreted as an extended lukewarm phase with a subdued glacial period (MIS 14), potentially contributing to the collapse of the WAIS. The concentration of $^{10}Be_{reac}$ and the $(^{10}Be/^{9}Be)_{reac}$ values show a notable increase following MIS 16, particularly

during MIS 14-11, supporting a lukewarm condition. This observation aligns with the evidence of enhanced sorting and higher silt percentage during this interval, indicative of increased meltwater discharge. The period between 550 and 250 ka shows TOC/TN levels within the range observed in the euphotic layer of the Ross Sea. $Ba_{ex}$ values are relatively higher during this interval, correlating with TOC levels. Therefore, the increased TOC peaks during this period are likely due to higher productivity in a lukewarm condition.

The late Pleistocene period is characterized by coarser grain size, IRD-rich layers, and high-frequency variations in MS values. The $(^{10}Be/^{9}Be)_{reac}$ increases significantly after 250 ka , showing opposite trend to the MS values. The upwelling of CDW contributed to the increase of the $(^{10}Be/^{9}Be)_{reac}$ in this interval. The opposite trend between MS and the $(^{10}Be/^{9}Be)_{reac}$ supports the hypothesis that MS values are higher during the glacial periods. Post 250 ka period, Opal % shows a positive correlation with $(^{10}Be/^{9}Be)_{reac}$ and a weak positive trend with TOC, while displaying a negative trend with MS. This suggests increased

late Pleistocene productivity during warmer periods and vice versa.

**Data availability**

All data for this paper is available in both the main manuscript and supplementary files. Any missing data will be available upon request.

**Author contributions**

All authors approved the manuscript and agreed on its submission. CD and YBS conceptualized the study and conducted the field investigations with MKL, JIL, and KCY. KOPRI members (MKL, JIL, and KCY) were responsible for funding acquisition. CD, HHR, and BYU designed the $^{10}Be$ lab experiments with BYY. CD, AS, and KOPRI members performed all formal analysis and result interpretations. CD, YBS, and AS prepared the manuscript with contributions from all co-authors.


**Competing interests**

The authors declare that they have no conflicts of interest.

**Acknowledgements**

We are grateful to Purevmaa Khandsuren and Eunji Kim for managing chemical treatment of $^{10}Be$ samples.

**Financial support.**

This research was funded by Ministry of Oceans and Fisheries (PE24090).



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
