# Peer review of "Drastic changes in Depositional Environments at the Ross Sea Continental Margin since the Mid-Pleistocene: More evidence for West Antarctic Ice Sheet collapse"

_Climate of the Past, 2024_

## Referee Comment (RC2)

The authors present a valuable dataset from the Ross Sea region that offers insights into the dynamics of the West Antarctic Ice Sheet over the past ~780,000 years. While they provide a substantial amount of data, there are several issues that need to be addressed.

1. General Comments

1.1 Age Model:

In Lines 90-99, the authors state that the age model for RS15-LC47 is based on correlating magnetic susceptibility (MS) to a nearby core referenced in Bollen et al. (2022). They assert a "noteworthy" similarity between their MS records and those from Bollen et al. (2022) in Line 119 and mention a "comprehensive approach utilizing 12 tie points" (Line 114) to align their records with the Bollen et al. (2022) record. However, the authors do not clearly demonstrate how they tested this "similarity" or explain the methods and rationale behind correlating the MS records.

For instance, in Figure 2C, the patterns older than 250 ka, according to the current age model, appear quite different. Bollen et al. (2022) show a relatively muted signal during ~300-550 ka, yet the LC42 record still displays numerous high-frequency peaks. Given the potential for bioturbation and hiatus/event layers in LC47, how can the authors confidently assert that these correlations are "robust"?

This issue is fundamental since an unreliable age constraint undermines the entire discussion. The authors are encouraged to meticulously re-examine their sedimentation records, incorporating more biostratigraphic controls as in Bollen et al. (2022), before making further interpretations.

2. Typos and Other Comments:

2.1 The manuscript frequently repeats full names and acronyms. For instance, "Middle-Pleistocene Transition (MPT)" is repeated in Lines 50, 229, 399, 449, and 457, and numerous other terms (AABW, MS, CDW, ASC, etc.) are also redundantly mentioned. The authors should carefully review the manuscript for such repetitions before resubmission.

2.2 The discussion on the MPT event is also perplexing since the record in this manuscript only covers a very brief portion of the later part of the MPT. The authors should consider comparing their records with well-dated ice core records instead. Although the introduction suggests a lack of records for the past few hundred thousand years, especially in the Ross Sea, many high-quality records exist that are not referenced in this manuscript.

2.3 Another point of confusion is the extensive discussion on the importance of Be isotope measurement and principles in Lines 65-75. By the end of the manuscript, Be isotopes do not seem to play a significant role in the model the authors propose, instead serving a supportive role that requires support from other evidence.

2.4 The correlation presented in Figure 6 lacks clarity and statistical robustness due to the limited data points for each time period. The authors should consider grouping some of the time periods and providing clear descriptive statistical tests to support their arguments. A similar issue is present in Figure 3, where there is no clear evidence indicating which correlations are strong and which are not.

Finally, Figure 4 is difficult to read due to an overload of poorly organized information, and the color labeling is inconsistent with Figure 2.

---

## Author Comment (AC1)

**Reviewer # 1**

**Comment:** The title overstates what the manuscript is able to demonstrate in its current form.. or perhaps in general.

**Reply:** Thank you for highlighting this issue. We have revised the title to: **"Drastic Changes in Depositional Environments at the Ross Sea Continental Margin Since the Mid-Pleistocene."**

**Comment:** A lot of weight has ben placed on interpretation of the magnetic susceptibility but the assumptions made are incorrect. MS is a magnetic mineral concentration proxy and will provide insights into the ratios of ferri/ferro magnetic, paramagnetic and diamagnetic components in a sample. Ohneiser et al 2019 showed the relationship very clearly MS and magnetic mineral concentration at LC47.

**Reply:** We have rectified our assumptions on the interpretation of MS values. Please see line no. 123-133. **"Ohneiser et al. (2019) provide a comprehensive analysis of the magnetic mineralogy of sediment cores from the Ross Sea. Their study shows that sediment cores collected near the continental margin are relatively coarser and contain magnetic minerals with lower coercivity compared to those from deeper waters. Additionally, there seems to be a connection between increased concentrations of magnetic minerals and glacial periods, as indicated by the benthic $\delta^{18}O$ stack LR04 (Lisiecki and Raymo, 2005). The elevated levels of magnetic minerals during these glacial periods suggest changes in sediment transport or winnowing processes, which are associated with variations in ice volume and bottom water flow from the continental shelf.."**

**Comment:** You need to reference the Ohneiser et al 2019 paper which provides the chronology for LC47. The B/M boundary at 11.65 m (773 ka) provides your chronological anchor from which you can correlate to LC42.

**Reply:** We have referred to Ohneiser et al. (2019) paper in line no. 101-103.

**Comment:** Carefully read the Bollen et al. (2022) paper and see how their data and interpretations work with yours. In their work they develop a more complex chronology of ice advance and retreat than what is presented for LC47 which makes me think that some more detail can be extracted form this record.

Reply: Thank you for your suggestions. We have added additional paleoclimatic inferences, referencing Bollen et al. (2022), which can be found in Sections 4.1.3 and 4.3.2. However, due to the lack of a well-established chronology, we could not incorporate high resolution climatic variations.

**Comment:** There appears to be discussion in this manuscript on lower portions of the core than what is presented.

**Reply.** We partially agree with the reviewer's observation. The lower section received more detailed discussion due to the interesting values of sedimentary components observed between 773 and 550 ka. However, we have now expanded

our interpretations in the other sections as well. Please refer to Sections 4.1.3 and 4.3.2 for these additional insights.

**Comment:** Line fits and statistics need to be presented so that we can see how strong the correlations are and in turn how valid the discussion and conclusions are.

**Reply:** Thanks for pointing the issue. Now, we have added line fits and statistics to all our figures.

**Comment:** Add MIS labels to all figures so we can see what is being referred to.

**Reply:** The suggestion has been included to the figures. Please see figs. 4 and 8.

**Comments:** There is no mention of MIS 12. This is a deep cold glacial which some speculate resulted in reorganisation in Antarctic more persistent sea-ice cover.

**Reply.** We appreciate the esteemed reviewer's concerns regarding MIS 12. However, we did not observe any noticeable signatures of this interval in our sediment core, nor has it been clearly identified in the nearby sediment core RS15-LC42. This period corresponds to a zone of dampened MS values in the Ross Sea sediment core. We speculate that the absence of this signature in our sedimentological and geochemical results may be due to the core's location being farther from the Antarctic continental margin.

**Comment:** The discussion on CaCO3 preservation needs more development. Bonaccorsi et al., 2007 present a record of only the last few thousand years so this is not comparable with the LC47. No mention is made to carbonate production and its relationship to LC47. What is the carbonate? Is it foraminifera, coccoliths? Is it old recycled washed in from the shelf or new carbonate? Can assumptions on dilution made from CaCo3 data be corroborated with microscope images demonstrating varying dissolution? To discuss carbonate preservation more data are needed and a discussion on assumptions of production rates which will have been strictly during warm interglacials.

**Reply:** We agree that our data set is insufficient to discuss the calcium carbonate preservation elaborately. However, following the reviewer's concerns we have modified the discussion on the carbonate preservation. Please see section 4.3.2.

**Comment:** The Be data discussion needs to be reviewed by a specialist other than me. Figure 6 needs some statistics to see how good the line fits are so the reader can deduce how robust the discussion and conclusions are. At the moment I can't tell if the data support the discussion or conclusions. More work is needed in clear data presentation and their statistical significance before discussions on CDW/ACC and ice sheet evolution can be evaluated.

**Reply:** We agree that our data needed statistical analysis done. We have added line fits and statistics to all our figures.

**Line wise comments:**

Lines 24-25. From these data you can't propose a collapse of the WAIS or ice shelf collapse.

We agree with the reviewer and removed the WAIS collapse statement from our abstract.

Line 39. You need a reference to support the statement 'Ross Sea is the primary drainage...'

Reference added to the specified section. Please check line no. 41.

Line 42: What is meant by elucidating historical ice sheet dynamics? Ice sheet dynamics which have happened since the arrival of humans and documentation? Current ice sheet dynamics?

We have modified the lines: "Thus, studying the sedimentation pattern in this region is critical in elucidating paleoceanographic changes and past ice sheet dynamics". Please see line no. 50-51.

Line 54: Naish et al do not present organic geochemical data.

Thanks for the comment. We have modified the lines as " Sedimentary facies records from AND-1B showed a consistent increase in the duration and the coverage of sea ice during the past 3 Ma (Naish et al., 2009)". Please see line no. 66-67.

Line 55: Suggest referring to the Ohneiser et al 2023 Nat Geo paper here where we showed persistent ice advance and retreat paced by obliquity until 400 ka. Its the only other Pleistocene record from on the shelf which spans the last c 1 myr.

We have referred to the findings from Ohneiser et al. (2023) in line no. 56-58.

Line 97: You need to reference Ohneiser et al 2019. Ohneiser et al established the magnetostratigraphy and the placed the B-M boundary at 11.65 m (773 Ka) which provides the original age model for LC47 (and LC42). Bollen et al 2022 developed Ohneiser Line 115. This age has been revised and is now 773 ka.

We have revised the B-M boundary age based on the work of Ohneiser et al. (2019). Please see the updated text in lines 101-104: Ohneiser et al. (2019) utilized magneto-biostratigraphic age models, integrating diatom records and geomagnetic reversals, to establish a chronology for sediment cores from the Ross Sea. In the RS15-LC47 core (this study), the Brunhes-Matuyama (B-M) reversal, dated at 773 ka, was identified at a depth of 1165 cm. Our study focuses up to the B-M reversal (773 ka), which is the only established magneto-biostratigraphic age marker for the RS15-LC47 core.

Line 120: You need to show the statistics associated with the line fits to see how strong the correlation is.

Yes, we have added line fits to all relevant figures. Although the R² values for some line fits are not particularly strong, there are still distinctive trends between the parameters. While the correlation coefficients may not always be high, the observed trends in the line fits offer valuable insights into the depositional processes and environmental changes along the Antarctic margin. We believe these trends are meaningful and reflect the inherent variability of the depositional environment.

LIne 120: I suggest making cross plots of MS vs biogenic opal and other parameters. I suspect the correlation biogenic % and Ms will be stronger then grainsize and MS.

We have followed the suggestion and plotted MS versus biogenic opal (Fig. 7b). Please refer to lines 387-391: "Opal content does not correlate with MS values (Fig. 7b), likely because Opal content lacks the high-frequency variations observed in MS in the studied core. In general, Opal$_{MAR}$ is influenced by opal deposition and their preservation, while opal content may be additionally affected by dilution with other sedimentary components, primarily lithogenic particles drift sediments.".

Line 125: Ohneiser et al 2023 showed that the LR04 does not reflect ice advance and retreat pacing in the Ross Sea. δ18O is a globally mixed record and not region specific.

We have referred to Ohneiser et al. (2023) in lines 130-133. "

**This lack of correlation may be attributed to the asynchronous nature of glaciation worldwide, as highlighted by Raymo et al. (2006). Consequently, this global signal in δ$^{18}$O may not accurately capture specific ice volume changes in the Ross Sea region or the expansion/retreat of the WAIS (Ohneiser et al., 2023). The studied core RS15-LC47 is too far from the ice margin and will capture a mixed oceanographic and ice sheet advance record".**

Line 125:You need to acknowledge here that LC47 may be too far from the ice margin and will capture a mixed oceanographic and ice sheet advance record.

We have acknowledged this fact. Please see the above reply.

Line 140: Strictly speaking these are no organic geochemical proxies since organic geochemistry implies analysis of carbo base organic molecules (alkenones, alkenes/anes.. other lipids etc etc).

We agree and have modified the term organic geochemical proxies section as geochemical proxies. Please see section 3.2.

Line 149. It would be good to demonstrate whether the visual correlation is real.. i.e. statistically.

We have done DTW analysis to the MS values of two cores to test their similarity (Supplementary material I). DTW is a robust statistical technique used to measure the similarity between two time series by optimally aligning their temporal sequences, accounting for potential shifts and variations along the time axis. This method is particularly suited to our data, as it allows for the alignment of similar trends even when minor temporal discrepancies exist, which is expected due to natural variability between cores in close proximity.

LIne 164. Need to provide a reference on where these broad climatic zones come from.

We have modified the lines. Please see caption of fig. 4. **"The broad climatic zones based on sedimentological and geochemical analysis are represented with coloured boxes".**

LIne 170. refer to figure.

We have referred to the figure. Please see line no. 164.

LIne 189 - poor sorting?

We have included the suggestion. Please see line no. 179.

Line 193 - Ohneiser et al 2019 interpreted these thin gravel as lag surfaces which indicate enhanced current strength and winnowing. They likely provide a protective armoured surface once formed that prevents further erosion of underlying sediments.

We have added this interpretation. Thanks! Please see lines 183-185.

LIne 197 - what is meant by 'this temporal span'? this interval?

We have included the suggestion. Please line no. 190.

LInes 205-215. This reads like a collection of bullet points. It should be revised into a coherent block of text with references.

We have modified this section Please see lines 190 to 207. " This  period is characterized by large scale expansion of the Antarctic ice sheets (McClymont et al., 2013; Sutter et al., 2019)  and northward migration of the polar front (Kemp et al., 2010). During periods of full glaciation, the ice sheet extends to the outer continental shelf, restricting the Antarctic Slope Current (ASC) to a narrower slope (Conte et al., 2021). This confinement limits energy dissipation and water exchanges, potentially intensifying the ASC, which plays a crucial role in regulating Circumpolar Deep Water (CDW) inflow and Ross Sea Bottom Water (RSBW) outflow. During full glacial intervals, the expansion of the ice sheet across the Antarctic continental shelf is evident from the erosional surfaces in the sedimentary strata (McKay et al., 2012b; Bollen et al., 2022). In the Ross Sea sector, there have been multiple instances of the grounding line advancing toward the shelf break during glaciations since the Mid-Pleistocene Transition (MPT) (McKay et al., 2012b). The advance of the grounding line initiates sediment gravity flows and turbid plumes, which are funneled to the outer continental margin through the canyon networks (Fig. 5c; Larter and Barker, 1989; Lucchi et al., 2002). The presence of millimeter-scale faults observed within this interval indicates they are synsedimentary, likely forming due to the increased sedimentation associated with these turbid plumes. As the finer fractions of these flows settle, they are transported westward by along-slope currents, highlighting the combined influence of downslope sediment gravity flows and along-slope currents in shaping channel-levee systems along the Antarctic margin during glacial periods. This sedimentation pattern is further corroborated by the higher clay content, mass accumulation rates (MAR), and poor sorting observed at the study site during this interval (Fig. 4). Additionally, the increased sedimentation rates and harsh glacial conditions likely suppressed biological activity. The elevated and irregular magnetic susceptibility (MS) values observed during this period also suggest poor sorting, consistent with sedimentation dominated by gravity flows."

Line 214. McKay et al 2012 do not discuss winnowing of magnetic susceptibility.

Thanks for the suggestions. We have removed this sentence from the paragraph (Above paragraph).

LIne 260. Im not sure this is a correct statement. The AND-1B site is in a very different setting because it was grounding zone proximal with multiple grounding zone advance and retreat cycles directly over the site. LC47 is oceanic and has never been near the grounding zone.

Thanks for the suggestion. We have modified the section 4.1.3.

LIne 261 - where did temperatures increase by 1-2C°?

We have included the suggestion in the line no. 240-250. Ice sheet models and proxy records from the Antarctic continental margin indicate that, following MIS 11, interglacial temperatures increased by 1-2°C in the high latitudes of the Southern Ocean and Antarctica, leading to greater ice volume variability with smaller interglacial and larger glacial ice sheets (Sutter et al., 2019).

LInes 266-335- this discussion on Be data needs to be reviewed by someone other than me. Figure 6 needs some statistics to see how good the line fits are so the reader can deduce how robust the discussion and conclusions are.

We have added statistics to Fig. 6.

Lines 338 - most agree that the MPT was complete at around 800 kyr.

Please see below in the response of line 395

LInes 340 etc. the presented interval in LC47 does not extend older than MIS19!

Please see below in the response of line 395

Line 395. The record presented here does not cover the mid-pleistocene transition which is generally agreed to be complete at around 800 kyr. Please revise this section.

We have removed the discussion on MPT from the manuscript. Please see section 4.1.

---

## Author Comment (AC2)

**Reviewer # 2**

**Comment:** The authors present a valuable dataset from the Ross Sea region that offers insights into the dynamics of the West Antarctic Ice Sheet over the past ~780,000 years. While they provide a substantial amount of data, there are several issues that need to be addressed.

1. General Comments

1.1 Age Model:

In Lines 90-99, the authors state that the age model for RS15-LC47 is based on correlating magnetic susceptibility (MS) to a nearby core referenced in Bollen et al. (2022). They assert a "noteworthy" similarity between their MS records and those from Bollen et al. (2022) in Line 119 and mention a "comprehensive approach utilizing 12 tie points" (Line 114) to align their records with the Bollen et al. (2022) record. However, the authors do not clearly demonstrate how they tested this "similarity" or explain the methods and rationale behind correlating the MS records. For instance, in Figure 2C, the patterns older than 250 ka, according to the current age model, appear quite different. Bollen et al. (2022) show a relatively muted signal during ~300-550 ka, yet the LC42 record still displays numerous high-frequency peaks. Given the potential for bioturbation and hiatus/event layers in LC47, how can the authors confidently assert that these correlations are "robust"? This issue is fundamental since an unreliable age constraint undermines the entire discussion. The authors are encouraged to meticulously re-examine their sedimentation records, incorporating more biostratigraphic controls as in Bollen et al. (2022), before making further interpretations.

**Reply:** We appreciate the reviewer's concerns regarding the age model. In our study, we compare the magnetic susceptibility (MS) records of the sediment core with those from a nearby core, which has an established magneto-biostratigraphic age model (Bollen et al., 2022). Previous research on core RS15-LC42 by Bollen et al. (2020) indicates that the environmental settings of the Antarctic continental margin are not always synchronous with global events. Additionally, we compared our records with the EDC δD record (Jouzel et al., 2007) (Fig. 2), which did not yield consistent results.

For our study, we rely on the following age markers:

1. The Brunhes-Matuyama (B-M) boundary of the studied core, previously established magneto-biostratigraphically by Ohneiser et al. (2019).

2. The dampened MS values observed between 550-250 ka in sediment cores from the Ross Sea (Ohneiser et al., 2019).

These age markers are well-established and consistently observed across sediment cores from the Ross Sea, providing confidence in their accuracy. We used the established ages as tie points for age correlation, focusing on broad time intervals: 750-550 ka, 550-250 ka, and 250 ka to the present. Our results do not aim to reconstruct high-resolution paleoenvironmental changes; instead, they emphasize

broader temporal trends. Therefore, our interpretations are unlikely to be significantly affected by small chronological uncertainties. The studied intervals are clearly defined by the established age markers, and previous studies on the nearby sediment core (Bollen et al., 2022) do not indicate any erosional hiatus within these time frames. Thus, we are confident that our interpretations are robust against potential chronological errors that could affect the conclusions.

In addition, We have done DTW analysis to the MS values of two cores to test their their similarity (Supplementary material I). DTW is a robust statistical technique used to measure the similarity between two time series by optimally aligning their temporal sequences, accounting for potential shifts and variations along the time axis. This method is particularly suited to our data, as it allows for the alignment of similar trends even when minor temporal discrepancies exist, which is expected due to natural variability between cores in close proximity. The DTW analysis shows that the MS values of two cores have similar patterns, with the DTW distance of 184.76 indicating a reasonable degree of similarity after accounting for temporal or local shifts.

1. Typos and Other Comments:

2.1 The manuscript frequently repeats full names and acronyms. For instance, "Middle-Pleistocene Transition (MPT)" is repeated in Lines 50, 229, 399, 449, and 457, and numerous other terms (AABW, MS, CDW, ASC, etc.) are also redundantly mentioned. The authors should carefully review the manuscript for such repetitions before resubmission.

**Reply:** Thanks for the comment. We have taken care of this. The revised manuscript does not repeat full names and acronyms.

2.2 The discussion on the MPT event is also perplexing since the record in this manuscript only covers a very brief portion of the later part of the MPT. The authors should consider comparing their records with well-dated ice core records instead. Although the introduction suggests a lack of records for the past few hundred thousand years, especially in the Ross Sea, many high-quality records exist that are not referenced in this manuscript.

**Reply:** Thanks for the comment. We have compared our results with that of Ohneiser et al. (2019), Ohneiser et al. (2023) and Hillenbrand et al. (2009). Which are some of the well dated sediment records from the region.

2.3 Another point of confusion is the extensive discussion on the importance of Be isotope measurement and principles in Lines 65-75. By the end of the manuscript, Be isotopes do not seem to play a significant role in the model the authors propose, instead serving a supportive role that requires support from other evidence.

**Reply:** Thank you for your valuable feedback. We appreciate your concerns regarding the role of Be isotopes in our manuscript. The Be isotope measurements were crucial in interpreting the paleoenvironmental conditions of the studied time interval, particularly in understanding the depositional environments of biogenic sedimentary components. While we acknowledge that different proxies capture distinct environmental signatures, the Be isotope analysis provided critical confirmation of the

environmental conditions under which various sedimentary components were deposited. For instance, the results from Be isotope analysis were instrumental in interpreting the organic geochemical components, offering insights that sedimentological analyses alone could not fully resolve. Be isotopes provided a higher resolution of paleoenvironmental signals, which significantly enhanced our understanding of the depositional dynamics. Notably, Section 4.3, "Production and Preservation of Biogenic Material at the Ross Sea Continental Margin," heavily relies on interpretations derived from Be isotope data, demonstrating their key role in our study. While Be isotopes may appear supportive, they underpin critical interpretations throughout the manuscript, especially in contexts where other evidence alone would not suffice. We have revised the manuscript to better emphasize this pivotal role, ensuring that the integration of Be isotopes into our model is clear and well-justified.

2.4 The correlation presented in Figure 6 lacks clarity and statistical robustness due to the limited data points for each time period. The authors should consider grouping some of the time periods and providing clear descriptive statistical tests to support their arguments. A similar issue is present in Figure 3, where there is no clear evidence indicating which correlations are strong and which are not.

**Reply:** Thank you for your thoughtful feedback. In response to your comments, we have added statistical data to all relevant figures. For Figure 6, we focused on analyzing the slopes of the bivariate plots rather than relying solely on correlation coefficients. This approach allowed us to better capture the relationship between the variables over different time periods. Most of the Be isotope groups presented in the bivariate plots demonstrate strong correlations, with the exception of the longer records for the 250 ka – Present and 773 ka – Present intervals. These longer records tend to provide more generalized slopes, which are less precise compared to shorter intervals that exhibit better correlation and more defined trends. Regarding the 250 ka – Present timeframe, we were unable to divide this period into distinct subgroups because the Be isotope data for each subgroup were too limited to generate statistically robust slopes and trendlines.

Finally, Figure 4 is difficult to read due to an overload of poorly organized information, and the color labeling is inconsistent with Figure 2.

**Reply:** Thank you for your feedback on Figure 4. To improve the clarity and readability of the data, we have divided the original Figure 4 into two separate figures: Figure 4 and Figure 8. This separation allows for a more organized presentation of the information, enhancing overall quality and interpretability. Additionally, we have refined the cartography for better visual representation and ensured that the color labeling is now consistent. These adjustments should make the figures more accessible and easier to compare across the manuscript.